# Causal Induction from Visual Observations for Goal Directed Tasks

## Abstract

Causal reasoning has been an indispensable capability for humans and other intelligent animals to interact with the physical world. In this work, we propose to endow an artificial agent with the capability of causal reasoning for completing goal-directed tasks. We develop learning-based approaches to inducing causal knowledge in the form of directed acyclic graphs, which can be used to contextualize a learned goal-conditional policy to perform tasks in novel environments with latent causal structures. We leverage attention mechanisms in our causal induction model and goal-conditional policy, enabling us to incrementally generate the causal graph from the agent's visual observations and to selectively use the induced graph for determining actions. Our experiments show that our method effectively generalizes towards completing new tasks in novel environments with previously unseen causal structures.

## 1 Introduction

Causal reasoning is an integral part of natural intelligence. The capacity to reason about cause and effect has been observed in humans and other intelligent animals as a means of survival (Blaisdell et al., 2006; Taylor et al., 2008). Such capacity plays a crucial role for young children in their interaction with the physical world. As behavioral psychology studies have shown, young children discover the underlying causal mechanisms from their play with the world (Schulz & Bonawitz, 2007), and their knowledge of causality in turn facilitates their subsequent learning of objects, concepts, languages, and physics (Rehder, 2003; Corrigan & Denton, 1996).

Nowadays, data-centric methods in artificial intelligence, such as deep networks, have achieved tremendous success in learning associations between inputs and outputs from large amounts of data, such as images to class labels (He et al., 2016). However, empirical evidence indicates that the absence of correct causal modeling in these methods is one potential reason for a lack of generalization, causing image captioning models to generate unrealistic captions (Lake et al., 2017), deep reinforcement learning policies to fail in novel problem instances (Edmonds et al., 2018), and transfer learning models to adapt slower to new distributions (Bengio et al., 2019).

In this work, we propose to endow a learning-based interactive agent with the capacity of causal reasoning for completing goal-directed tasks in visual environments. Imagine that a household robot enters a new home for the first time. Without prior knowledge of the wiring configuration, it has to toggle the switches and sort out the correspondences between lights and switches, before it can be commanded to turn on the kitchen light or the bathroom light. We refer to the first phase of toggling switches as **causal induction**, where the agent discovers the latent cause and effect relations via performing actions and observing their outcomes; and we refer to the second phase of turning on specific lights as **causal inference**, where the agent uses the acquired causal relations to guide its actions for the completion of a task. To build an effective computational model for causal induction and inference, we have to address generalization towards novel causal relations and new task goals at the test time, both of which can be unseen during training.

We cast this as a meta-learning problem of two phases following Dasgupta et al. (2019). In the first stage, we use a **causal induction model** to construct a causal structure, i.e., a directed acyclic graph of random variables, from observational data from an agent's interventions. In the second stage, we use the causal structure to contextualize a **goal-conditional policy** to perform the task given a goal. However, in contrast to Dasgupta et al. (2019) we explicitly construct the causal structure instead of a

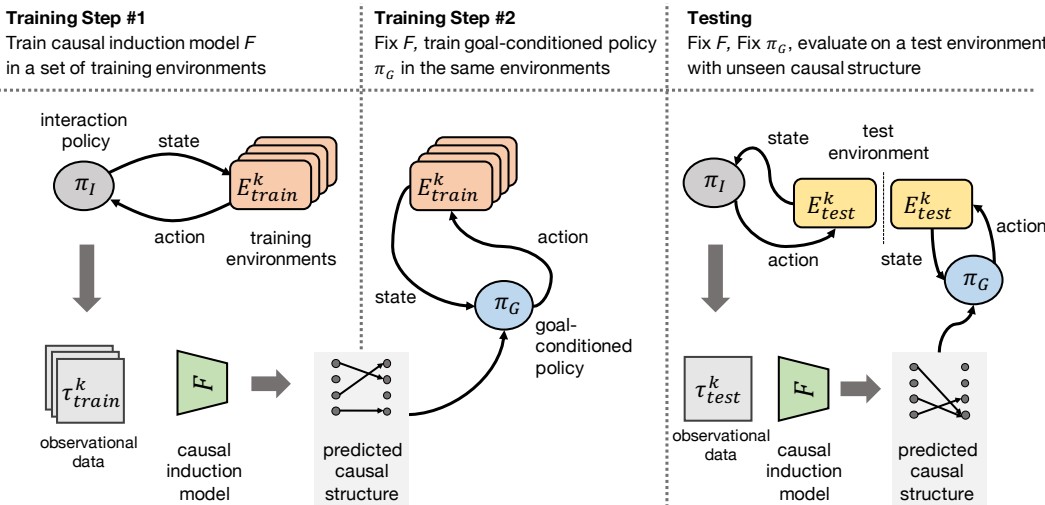

Figure 1: **Overview of Causal Induction and Inference Procedure.** During training each episode samples one of $K$ training environments and uses the interaction policy $\pi_I$ to probe the environment and collect a trajectory of visual observations. Using supervised learning we train the causal induction model $F$, which takes as input the trajectory of observational data and constructs $\hat{C}$, the estimate of $C_{train}^k$, which captures the underlying causal structure. Then, the predicted structure $\hat{C}$ is provided as input to the policy $\pi_G$ conditioned on goal $g$, which learns to use the causal model to efficiently complete a specified goal in the training environments. At test time, $F$ and $\pi_G$ are fixed and the agent is evaluated on new environments with unseen causal structures.

latent feature encoding, leading to substantially better generalization towards new problem instances in multi-step tasks as opposed to simplistic one-step querying.

To this end, we propose two technical contributions: 1) an iterative causal induction model with attention, which learns to incrementally update the predicted causal graph for each observed interaction in the environment, and 2) a goal-conditioned policy with an attention-based graph encoding, forcing it to focus on the relevant components of the causal graph at each step. We find that by factorizing the induction and inference processes through the use of causal graphs, it generalizes well to unseen causal structures given as few as 50 training causal structures. We compare our approach to using the ground-truth causal structure (which provides oracle performance), a non-iterative architecture which directly predicts the causal structure, and encoding the observation data into the LSTM (Hochreiter & Schmidhuber, 1997) memory of the policy similar to prior work (Dasgupta et al., 2019). We demonstrate that our method outperforms the baselines and achieves close to oracle performance in terms of both recovering the causal graph and success rate of completing goal-conditioned tasks, across several task sizes, types, and number of training causal structures.

## 2 PROBLEM STATEMENT

We formulate the agent's interaction in a Goal-Conditioned Markov Decision Process (MDP) defined by a six tuple $(\mathcal{S}, \mathcal{A}, p, \mathcal{G}, r, \gamma)$, where $\mathcal{S}$ is the state space, $\mathcal{A}$ is the action space, $p : \mathcal{S} \times \mathcal{A} \to \mathcal{S}$ defines the transition dynamics, $\mathcal{G}$ is the goal space, $r : \mathcal{S} \times \mathcal{A} \times \mathcal{G} \to \mathbb{R}$ is a reward function where $r(s, a, g)$ gives the one-step immediate reward conditioned on the goal $g \in \mathcal{G}$, and $\gamma$ is the discount factor. Our goal is to learn a goal-conditioned policy $\pi_G : \mathcal{S} \times \mathcal{G} \to \mathcal{A}$ that maximizes the expected sum of rewards $R_t = \sum_{k=t}^{\infty} \gamma^{k-t} r(s_k, a_k, g)$.

In this work, we not only want $\pi_G$ to generalize well across goals in $\mathcal{G}$, but consider a more ambitious aim of making $\pi_G$ generalize across a set of MDPs. We consider $\mathcal{M} = \{M^{(1)}, M^{(2)}, \ldots, M^{(K)}\}$ as the entire set of $K$ MDPs with the same state space and action space but different transition dynamics, where $M^{(k)}$ is defined by $(\mathcal{S}, \mathcal{A}, p^{(k)}, \mathcal{G}, r, \gamma)$. The dynamics $p^{(k)}$ determines underlying causal relations between states and actions. Taking the same action at the same state could lead to a different next state under a different dynamics. We expect our agent to operate on its first-person vision and has no access to the latent causal relations. It receives high-dimensional RGB observations and has

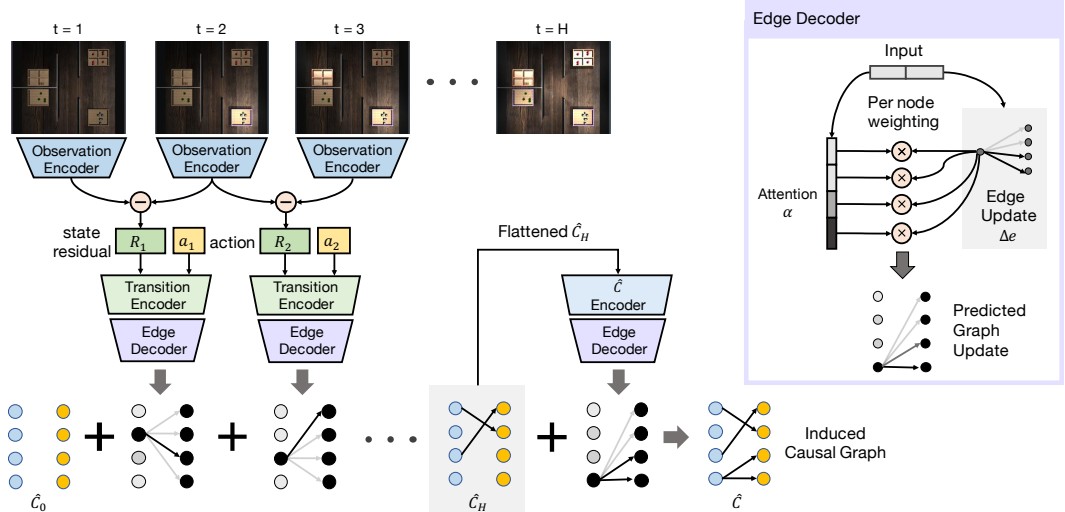

Figure 2: **Iterative Causal Induction Network**. Our iterative network architecture for inducing the causal structure from a visual trajectory of observational data with horizon $H$. First each frame is encoded into a latent state embedding $s$. Then the difference between state embeddings across time steps (state residual) is computed, and concatenated with the corresponding action. This is fed into the Edge Decoder module, which predicts an edge update, as well as an attention vector which is used to weight how the edge update is applied to nodes. This results in an estimate of the causal graph $\hat{C}_i$ at the $i^{th}$ iteration. On the last step one more edge update based on the current graph is applied, and a final predicted graph $\hat{C}$ is outputted.

to induce a causal model from observational data. As illustrated in Figure 1, the overall procedure has two stages: 1) we execute a heuristic interaction policy $\pi_I$ to collect a sequence of transitions $\tau = \{(s_1, a_1), (s_2, a_2), \ldots\}$, which is consumed by an induction model $F$ to construct a latent causal model $\hat{C} = F(\tau)$; and 2) we use the causal model $\hat{C}$ to contextualize a goal-conditioned policy $\pi_G$ such that it can perform tasks in the new MDP with novel causal relations. In this work we focus on causal graphs $\hat{C}$ which are bipartite with a fixed set of "cause" variables mapping to a fixed set of "effect" variables. We formulate this as a meta-learning problem (Dasgupta et al., 2019; Finn et al., 2017). We partition the set of all MDPs $\mathcal{M}$ into two disjoint sets $\mathcal{M}_{train}$ and $\mathcal{M}_{test}$. During training, we learn our induction model $F$ and goal-conditioned policy $\pi_G$ with $\mathcal{M}_{train}$. During testing, we evaluate whether $F$ is able to learn from the observational data from $\pi_I$ in a novel MDP from $\mathcal{M}_{test}$ to construct a causal model that can be used by $\pi_G$ to perform tasks in this new MDP.

Direct modeling of causal relations in raw pixel space is intractable due to the large dimensionality. Following Chalupka et al. (2014), we assume that cause and effect in our problems can be defined on a handful of causal macro-variables. For example, the illuminance of the kitchen (the agent's visual observation) is determined by the on and off of the kitchen light (macro-variable), which is caused by toggling the state of the switch that controls the light (another macro-variable). This assumption enables us to construct a directed acyclic causal model $\hat{C}$ to represent the causal effects of actions on these macro-variables. Given the set of macro-variables, the induction model $F$ predicts directed edges between them from visual observations. A primary challenge here is the confounders raised from partial observability and spurious correlations in the agent's visual perception. For example, illuminance changes in the kitchen might be due to turning on/off the kitchen light or the living room light. Hence, it requires the agent to disentangle the correct causal relations from visual inputs.

## 3 METHOD

The goal of our method is to enable a policy to complete goal-conditioned vision-based control tasks in environments with unseen causal structures, given only a short trajectory of observational data in the environment. Prior work (Dasgupta et al., 2019) has shown promising results on simplistic one-step querying problems using an LSTM-based policy which encodes the interaction into the policy's memory. Our hypothesis is that to generalize in complex multi-step control problems, a more

structured induction and policy scheme will be required. To address this, we propose iterative updates and attention bottlenecks in the induction model $\hat{C} = F(\{(s_1, a_1), (s_2, a_2), \ldots\})$ and in the policy $a = \pi_G(s, g, \hat{C})$ respectively, which we demonstrate significantly improves generalization to unseen causal structures.

## 3.1 Iterative Causal Induction Network

Inducing the causal structure from raw sensory observations requires accurately capturing the unique effect of each action on the environment, while accounting for confounding effects of other actions. We hypothesize that the causal induction network that best generalizes will be one which disentangles individual actions and their corresponding effect, and only updates the relevant components of the causal graph.

We implement this idea in our iterative model, where we begin with an initial guess of the causal structure $\hat{C}$ which has all edge weights of 0 (meaning we assume no causal relationships). We then use an Observation Encoder $E$ to map each image of the observational data to an encoding $s$ and compute the state residual

$$R_i = E(s_{i+1}) - E(s_i) \tag{1}$$

between subsequent time steps $i$ and $i + 1$. This $R_i$, which captures the change in state is then concatenated with the corresponding action $a_i$, and then fed into the Edge Decoder module. The output of the Edge Decoder module is an update to the edge strengths of the causal graph $\Delta\hat{C}$. This update is applied to each observed transition, that is $\hat{C}_{t+1} = \Delta\hat{C} + \hat{C}_t$, and at the final layer the whole graph is encoded to do a final edge update before the causal graph is predicted (see Figure 2). The Edge Decoder takes either the encoded $R$ and $a$, or the encoded edge matrix of $\hat{C}_H$, and outputs a $1 \times N$ soft attention vector $\alpha$ and a $1 \times N$ change to the edge weights $\Delta e$, where $N$ is the number of actions in the environment. The attention vector $\alpha$ is used to weight which nodes in the causal graph the edge update $\Delta e$ should be applied to. Thus at each iteration the update amounts to:

$$\hat{C}_{t+1} = (\alpha^T \Delta e) + \hat{C}_t, \alpha = \phi(R_{t+1}, a_{t+1}) \tag{2}$$

where $\phi$ is the Transition Encoder, a fully connected module (see Appendix A for details). Using this attention mechanism further encourages the module to make independent updates, which we observe enables better generalization.

## 3.2 Learning Goal-Conditioned Policies

The objective of the policy is given an initial image $s_0$, a goal image $g$, and the predicted causal structure $\hat{C}$, reach the goal within $T$ time steps. Additionally, the policy $\pi_G(s, g, \hat{C})$ is a reactive one, and thus can only solve the goal-conditioned task if it can learn to use the predicted causal structure $\hat{C}$. That is — since the policy has no memory, it cannot learn to induce the graph internally during inference time, and thus must use the causal graph $\hat{C}$.

We hypothesize that like the causal induction model, the policy which best generalizes is one which learns to focus exclusively on the edges in the causal graph which are relevant to the current step of the task. To that end we propose an attention bottleneck in the graph encoding, which encourages the policy to select edges pertaining to one "effect" at each step, which enables better generalization.

Specifically, the policy encodes the current image $s$ and goal image $g$. Based on this encoding it outputs an attention vector $\alpha$ of size $1 \times N$ over the "effects" in the causal graph. This vector is used to perform a weighted sum over the outputs of the $N \times N$ causal graph ($N$ causes, $N$ effects, and edges between them), resulting in a size $N$ vector of the selected edges $e$. The selected edges and visual encodings are used to output the final action:

$$a = \phi_3(E_\pi(s, g), \phi_2(e)), e = \hat{C} \cdot \alpha^T, \alpha = \phi_1(E_\pi(s, g)) \tag{3}$$

where $E_\pi$ has identical architecture to the image encoder $E$ in $F$, but which encodes both current and goal image and $\phi_i$ are all fully connected layers (see Figure 3).

## 3.3 Model Training

The induction network $F$ is trained using supervised learning in the limited set of training environments, specifically to minimize the $\ell_2$ reconstruction loss between the ground-truth causal graph $C$ and the predicted causal graph $\hat{C} = F(\{(s_1, a_1), (s_2, a_2), \ldots\})$.

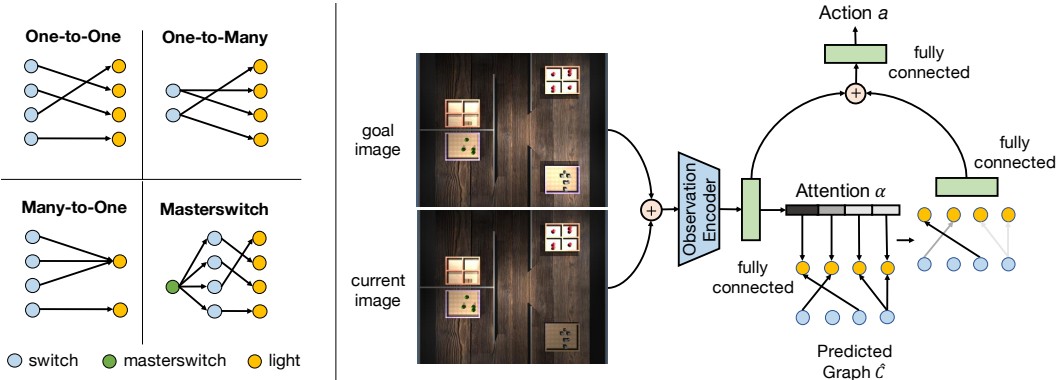

Figure 3: **Types of Causal Structures (Left)** We explore four types of causal structures, ONE-TO-ONE, ONE-TO-MANY, MANY-TO-ONE, and MASTERSWITCH. These cover a standard causal mapping, common cause causal patterns, common effect causal patterns, and causal chains. **Goal-Conditioned Policy (Right)**. The policy takes as input the current image, goal image, and predicted causal graph $\hat{C}$. The current image and goal image are concatenated channel wise and encoded. This encoding is used to predict an attention vector over the "effects" in $\hat{C}$ which extract the relevant edges, which is then concatenated with the image encoding to predict the action.

The policy $\pi_G$ is trained using the DAgger (Ross et al., 2010) algorithm by imitating a planner using the ground-truth causal graph in the training environments. Then $\pi_G$ is tested in unseen environments with only visual inputs and goal images. Specifically, in the training environments, the planner uses the ground-truth graph and privileged low dimensional state/goal information to compute an optimal plan to the goal, yielding a deterministic and uni-modal online expert. At each time step, the expert's action is added to the replay buffer of the policy, which is then trained using a standard cross entropy loss to imitate the expert given the current image and goal image. The policy is also injected with $\epsilon$-greedy noise during training, with $\epsilon = 0.3$. Network architectures and additional training details can be found in the appendix.

## 4 EXPERIMENTS

Through our experiments, we investigate three complementary questions: 1) Does our iterative induction network enable better causal graph induction?, 2) Does our attention bottleneck in the graph encoding in $\pi_G$ enable the policy to generalize better to unseen causal structures?, and 3) By combining our proposed $F$ and $\pi_G$, are we able to outperform the current state-of-the-art Dasgupta et al. (2019) on visual goal-directed tasks?

### 4.1 EXPERIMENTAL SETUP

**Task Definition:** We examine the multi-step task of light switch control. In particular an agent has control of $N$ switches (macro-variables) , which have some underlying causal structure of how they control $N$ lights (macro-variables). However, the macro-variables of the lights manifest themselves in noisy visual observations, whose partial observability and overlap result in confounding factors. The objective of the agent is starting from an initial state $s_0$, to control the switches to reach a specified lighting goal $g$, where both the state and goal are $32 \times 32 \times 3$ images. However as specified in the problem setup, the underlying causal structure is unknown to the agent, all that is provided is limited observational data from the environment. Thus the agent must (1) induce the causal structure between the switches and lights from the observational data, then (2) use it to reach the goal.

We explore 4 different types of causal patterns between the switches and lights (See Figure 3). The first type of causal structure is ONE-TO-ONE, in which each switch maps to one light. The second type of causal structure is MANY-TO-ONE (Common Effect (Keil, 2006)), where each switch controls one light, but multiple switches may control the same light. The third type is ONE-TO-MANY (Common Cause (Keil, 2006)), where all lights are controlled by at most one switch, but a single switch may control more than one light. Lastly, we also explore the MASTERSWITCH domain in which there is a Causal Chain (Keil, 2006) where only once one master switch is activated can the other switches causal effects be observed, applied on top of a ONE-TO-ONE causal structure.

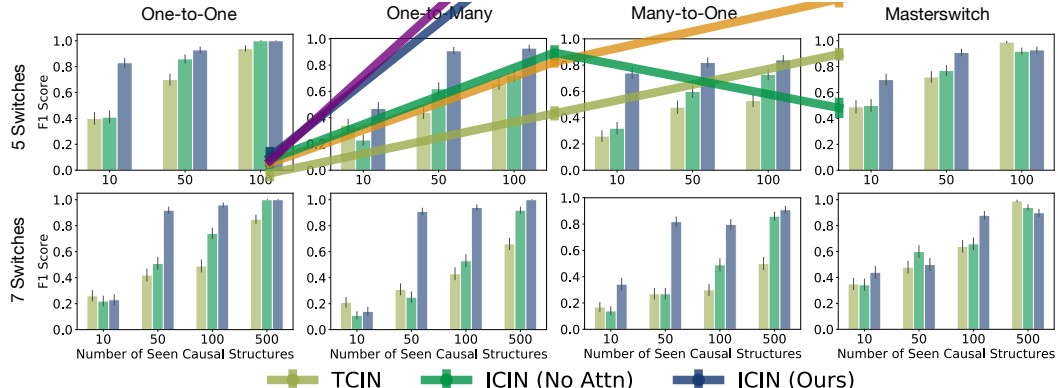

Figure 4: **F1 Score on Unseen Causal Structures**. The F1 Scores of edges on the predicted causal structure compared to the ground-truth on unseen causal structures. We compare across variable numbers of seen structures {10, 50, 100, 500} and problem size {5, 7}. Our iterative approach with attention outperforms the comparisons across almost all settings.

**Causal Graph:** We represent the causal structure $C$ as a bipartite graph with directed edges between $N$ switch variables and $M$ light variables in the environment, with edge strength corresponding to the likelihood that that switch controls the indicated light, with an additional $N$ edges in the MASTERSWITCH setting. In practice, we use $M = N$, so the space of possible edges is $N \times N$, but our proposed approach can be applied for arbitrary number of "cause" or "effect" variables.

**Environment Setup:** The trajectory of observational data from which we induce the causal structure is in visual space, consisting of a $32 \times 32 \times 3$ image as well as an size $A = N$ action vector for each timestep of the $H$ timestep trajectory. The visual scene consists of the $N$ lights mounted in a MuJoCo (Todorov et al., 2012) simulated house with 3 rooms, where the lights are mounted in the rooms and hallway, and the effect of each light is rendered onto the floor when they are turned on, which is captured by a top down camera. The illumination from the lights overlap, resulting in confounding factors, which must be disentangled by the model in order to correctly predict the causal structure. The state space (and goal space) of the policy are also $32 \times 32 \times 3$ images of the same environment and camera. The action space $A$ consists of size $N$, one discrete action for each switch. During policy learning, the goals are sampled uniformly from the space of possible lighting configurations under the environments causal structure, i.e., the set of all reachable states in the environment.

**Collecting Observational Data:** The observational data is collected using a heuristic interaction agent $\pi_I$, which executes a simple policy. In the all but the MASTERSWITCH case, the policy simply takes each action once (horizon $H = N$). In the MASTERSWITCH setting the exploratory agent presses each switch until one has an effect on the environment, and then proceeds to press each of the other switches (horizon up to $H = 2N - 1$).

## 4.2 EVALUATION METHODS

We evaluate the following methods and baselines to examine the effectiveness of our causal induction model and goal-conditioned policy.

First to examine how much our iterative causal induction network (**ICIN**) improves performance on inducing the causal graph we compare against a non-iterative induction model which uses temporal convolutions (**TCIN**), as well as an ablation of our method which uses an iterative model without the attention mechanism (**ICIN (No Attn)**). We compare these approaches based on F1 score of recovering the ground-truth causal graph.

Next, we compare the performance of the goal conditioned policy using all variants (ICIN, ICIN (No Attn), TCIN), compared to the previous work of Dasgupta et al. (2019) which induces the graph using the latent memory of the policy (**Memory**). Specifically, we provide the policy $\pi_G$ with LSTM memory, and before running DAgger, feed the interaction trajectory one step at a time through the policy. This end-to-end approach allows the causal graph to be encoded into the latent memory of the policy, to be used when doing a goal-directed task. We also compare to **Memory (RL/Low Dim)**, a version of Dasgupta et al. (2019) which has access to privileged state information (ground-truth

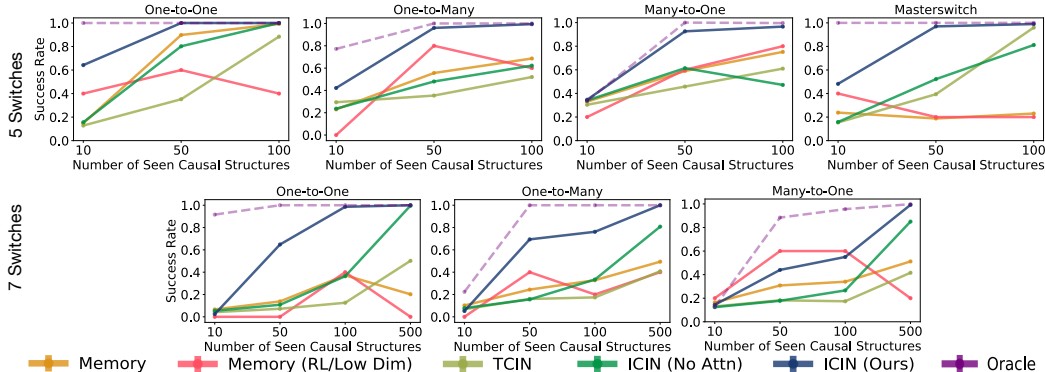

Figure 5: **Policy Success Rates (Unseen Causal Structures)**. The final success rates of the goal-conditioned policy for each method on unseen causal structures for either 10, 50, 100, or 500 seen causal structures for 5 or 7 switches. Our iterative approach achieves the best generalization on unseen tasks across almost all settings.

states), and is trained using model-free reinforcement learning with a dense reward. In this setting the same visual interaction trajectory is encoded into the policy's LSTM memory, but the actual policy receives a binary vector for state and goal and is trained using the PPO algorithm (Schulman et al., 2017). We also add a comparison to using the ground-truth causal graph (even at test time) as an Oracle (**Oracle**), which provides an upper bound on performance. All methods are compared based on success rates in unseen environments. Implementation details can be found in Appendix B.

Lastly, to understand how critical the attention bottleneck in the goal-conditioned policy is for generalization, we compare the success rates in unseen causal structures of the goal-condition policy using a graph induced by ICIN, with and without the attention bottleneck.

### 4.3 Causal Induction Evaluation

First we examine our approach's ability to induce the causal model from the trajectory of observational data. We report the F1 score (threshold=0.5) between edges of the predicted causal graph and ground-truth causal graph. We compute the F1 scores across up to 100 unseen tasks given 10, 50, 100, or 500 seen structures in the 5 and 7 switch problems (see Figure 4). We observe that across almost all settings, our iterative approach with attention (**ICIN**) dominates. While our method without attention generally outperforms the non-iterative baseline, both fall significantly behind our final approach, suggesting that the modularity that attention provides plays a large role in enabling generalization. Furthermore, we observe that our iterative attention approach especially outperforms the others when there are less training causal structures, which would suggest that by forcing the network to make independent updates, it is able to learn a general method for induction with limited training examples. This is likely because the attention forces the model to learn to update a single edge given a single observation, while being agnostic to the total graph, which is far less likely to overfit to the training structures. Qualitative examples of ICIN can be found in Appendix C.

### 4.4 Goal-Conditioned Policy Evaluation

We examine the success rate of the converged policy $\pi_G$ on 500 trials in unseen causal structures with randomized goals in Figure 5. We observe in most settings that the **Memory** based approach of Dasgupta et al. (2019) provides a strong baseline, outperforming the **TCIN** and **ICIN (No Attn)** baselines. We suspect that it learns that to best imitate the expert, it has to encode relevant information from the interaction trajectory into its latent memory, implicitly performing induction. While this works, it generalizes to unseen structures much worse than our proposed method, likely due to the compositional structure of our approach. The memory baseline which uses low dimensional states (**Memory (RL/Low Dim)**), and is trained via model-free RL also performs well, in fact beating our approach on a few cases in the 7 switch, MANY-TO-ONE setting. However in general the performance of this approach is much lower than ours, and likely the cases in which it does succeed can be attributed to its use of privileged information (ground-truth states) instead of visual

observations. In almost all cases **ICIN** significantly outperforms all baselines. In fact, in the 5 switch case our **ICIN** method nearly matches the Oracle, suggesting that it almost perfectly induces $C$.

Finally, we study the importance of our proposed attention bottleneck in the graph encoding in $\pi_G(s, g, \hat{C})$, which forces the policy to focus on only relevant edges at each timestep. We examine the success rate of using $\pi_G$ with the attention bottleneck compared to just flattening $\hat{C}$ given the current image, goal image, and predicted causal graph from ICIN.

We find that using the attention bottleneck in the graph encoder of $\pi_G$ yields a roughly 10% increase in success rate in the ONE-TO-ONE (1:1) and MASTERSWITCH (MS) cases, and a roughly 40% increase in success rate in the ONE-TO-MANY (1:K) and MANY-TO-ONE (K:1) cases. This is because by encouraging the policy to pick relevant edges, it has led to a modular policy which can 1) identify the changes it wants to make in the environment and 2) predict the necessary action based on the causal graph $\hat{C}$, enabling better generalization.

| $\pi_G$ | 1:1 | 1:K | K:1 | MS |
|---------|-----|-----|-----|-----|
| No Attn | 0.84 | 0.51 | 0.59 | 0.90 |
| Ours | **1.0** | **0.95** | **0.94** | **0.98** |

Table 1: **Policy Attention Bottleneck.** We observe in that across all settings using the attention bottleneck in the graph encoder of the policy significantly improves performance. Tested on 5 switches and 50 seen structures.

## 5 RELATED WORK

Causal reasoning has been extensively studied by a broad range of scientific disciplines, such as social sciences (Yee, 1996), medical sciences (Kuipers & Kassirer, 1984), and econometrics (Zellner, 1979). In causality literature, structural causal model (SCM) (Pearl, 2009) has offered a formal framework of modeling causation from statistical data and counterfactual reasoning. SCMs are a directed graph that represents causal relationships between random variables. Both causal induction (constructing SCMs from observational data) (Shimizu et al., 2006; Hoyer et al., 2009; Peters et al., 2014; Ortega & Braun, 2014) and causal inference (using SCMs to estimate causal effects) (Bareinboim et al., 2015; Bareinboim & Pearl, 2016) algorithms have been developed. Conventional methods have limited applicability in complex domains where observational data is high-dimensional and partially observable. Recent work has shown that causal learning can take advantage of the representational power of deep learning methods for inducing causal relationships from interventions (Dasgupta et al., 2019) and for improving policy learning via counterfactual reasoning (Buesing et al., 2018). However, they have focused on toy-sized problems with low-dimensional states. In contrast, our model induces and makes use of the causal structure for complex interactive tasks from raw image observations.

Lake et al. (2017) are among the first to discuss the limitations of state-of-the-art deep learning models in causal reasoning. There has been a growing amount of efforts in marrying the complementary strengths of deep learning and causal reasoning. Causal modeling has been explored in several contexts, including image classification (Chalupka et al., 2014), generative models (Kocaoglu et al., 2017), robot planning (Kurutach et al., 2018), policy learning (Buesing et al., 2018), and transfer learning (Bengio et al., 2019). Pioneer work on causal discovery with deep networks has applied to time series data in healthcare domains (Kale et al., 2015; Nauta et al., 2019). In addition, adversarial learning (Kalainathan et al., 2018), graph neural networks (Yu et al., 2019), and gradient-based DAG learning (Lachapelle et al., 2019) have been recently introduced to causal discovery, but largely focusing on small synthetic datasets. Most relevant to ours are Dasgupta et al. (2019) and Edmonds et al. (2018), which investigated causal reasoning in deep reinforcement learning agents. In contrast to them, our method directly learns an explicit causal structure from raw observations to solve multi-step, goal-conditioned tasks.

Generalization to new environments and new goals has been a central challenge for learning-based interactive agents. This problem has been previously studied in in the context of domain adaptation (Tzeng et al., 2015; Peng et al., 2017), system identification (Yu et al., 2017; Zhou et al., 2019), meta-learning (Finn et al., 2017; Sæmundsson et al., 2018), and multi-task learning (Andrychowicz et al., 2017; Schaul et al., 2015). These works have addressed variations in dynamics, visual appearances, and task rewards, while assuming fixed causal structures. Instead, we focus on changes in the latent causal relationships that determine the preconditions and effects of actions. We do so by leveraging attention mechanisms to focus only on relevant parts of the data, and idea also explored in prior work (Bengio, 2017).

## 6 CONCLUSIONS

We have proposed novel techniques for 1) causal induction from raw visual observations and 2) causal graph encoding for goal-conditioned policies, both of which lead to better generalization to unseen causal structures. Our key insight is that by leveraging iterative predictions and attention bottlenecks, it facilitates our causal induction model and goal-conditioned policy to focus on the relevant part of the causal graph. Using this approach we show better generalization towards novel problem instances than previous works with limited training causal structures.

In this work, we induce the causal structure from observational data collected by a heuristic policy. We plan to explore more complex tasks where probing the environment to discover the causal structure requires more sophisticated strategies, and develop algorithms that jointly learn the interaction policy.

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

# A  ARCHITECTURE DETAILS

## A.1  INDUCTION MODELS

### A.1.1  OBSERVATION ENCODER

The image encoder used in all models takes as input a $32 \times 32 \times 3$ image of the scene, feeds it through 3 convolutional layers with each followed by ReLU activation and $2 \times 2$ Max Pooling. The output filters of the convolutions are 8, 16, and 32 respectively, and the resulting $4 \times 4 \times 32$ tensor is mapped to latent vector of size $N$ using a single fully connected layer, where $N$ is the number of switches/lights.

### A.1.2  ICIN TRANSITION ENCODER

The transition encoder used in our iterative model takes as input a state residual of dimension $N$ and an action of size $N + 1$ concatenated together, and feeds it through fully connected layers of size 1024 and 512 each with ReLU activation, followed by another layer which outputs an attention vector of size $N$ (or $N + 1$ in the MASTERSWITCH case) with SoftMax activation and an edge update of size $N$ with Sigmoid activation. The first two layers are trained with dropout of 30%.

### A.1.3  ICIN $\hat{C}$ ENCODER

The causal graph encoder used in the last step our iterative model takes as input the $N \times N$ (or $N + N \times N$ in the MASTERSWITCH Case) flattened edge weights of the current graph $\hat{C}_H$ and feeds them though a single fully connected layer, which outputs an attention vector of size $N$ (or $N + 1$ in the MASTERSWITCH Case) with SoftMax activation and an edge update of size $N$ with Sigmoid activation.

### A.1.4  ICIN (NO ATTN)

The non-iterative ablation of our method has an identical architecture, but instead of the third fully connected layer outputting an attention vector of size $N$ with SoftMax activation and an edge update of size $N$ with Sigmoid activation, it instead updates a full set of $N \times N$ edge weights with Sigmoid activation.

### A.1.5  TCIN

The temporal convolution induction network uses the same image encoder as our approach. However the size $N$ state encodings are then concatenated with the size $N + 1$ action labels, and then are passed through three layers of temporal convolutions, with filter size [256, 128, 128] and a size 3 kernel with stride 1. The horizon $H$ by 128 dimensional output is then flattened and fed through fully connected layers of size 1024 and 512, each with 30% dropout. Finally, the size $N \times N$ causal graph is outputted with Sigmoid activation.

## A.2  POLICY ARCHITECTURE

### A.2.1  ATTENTION BASED $\pi_G$

The same image encoder as the induction models is used for the policy, except as input it takes a $32 \times 32 \times 6$ image which contains the current image and goal image, concatenated channel wise. The encoded image is then flattened and fed through a fully connected layer of size 128, which then outputs an attention vector of size $N$, which is used to do a weighted sum over the edges of the $N \times N$ causal graph, producing a vector of size $N$. This is then encoded to size 128, concatenated with the 128 dim image encoding, and def through 2 more fully connected layers of size 64 and ultimately outputting the final action prediction of size $N$.

In the non attention version the architecture is identical except the full graph is flattened, then encoded and concatenated directly with the image encoding.

## A.3  MEMORY BASELINE

In this baseline we use an image encoder as above, except there is an additional input for action. There is also an LSTM Cell of hidden dimension 256 which the image encoding and action encoding are fed into, which is then fed through fully connected layers of size 256, 64 which output the action.

### A.4   MEMORY (RL/LOW DIM) BASELINE

The policy is a MLP-LSTM policy as implemented in Hill et al. (2018), with two fully connected layers of size 64, and an LSTM layer with 256 hidden units. It is augmented with additional input heads for each step of the observational data, namely $32 \times 32 \times 3$ images and size $N$ actions.

## B   TRAINING DETAILS

### B.1   CAUSAL INDUCTION MODEL ($F$) TRAINING

Each causal induction model is trained for each split of seen/unseen causal structures as described in the experiments. The $F$ is trained offline on all $\tau_{train}^k$ and corresponding $C_{train}^k$, for 60000 training iterations using Adam optimizer (Kingma & Ba, 2015) with learning rate 0.0001 and batch size 512. They are implemented using PyTorch (Paszke et al., 2017) and trained on an NVIDIA Titan X GPU.

### B.2   DAGGER POLICY TRAINING

The policies trained with DAgger (Ross et al., 2010) are trained in the training environments with episode horizon $T = 2N$. The policy takes actions in the environment, and at each step an expert action is appended to the policy's memory buffer. The policy is then trained to imitate the experience in the memory. The expert uses the ground-truth causal graph and ground-truth low dimensional states to compute the difference between the goal and current state, and based on the graph what action needs to be taken. Each policy is trained for 100000 episodes, with learning rate 0.0001 and batch size from the memory of 32.

### B.3   RL POLICY TRAINING

The Memory (RL/Low Dim) baseline is trained using the Proximal Policy Optimization algorithm (Schulman et al., 2017) as implemented in Stable-Baselines (Hill et al., 2018). They use hyper-parameters $\gamma = 0.99$, 128 steps per update, entropy coefficient 0.01, learning rate 0.00025, value function coefficient 0.0001 and $\lambda = 0.95$. The policy itself consists of two fully connected layers of size 64, as well as an LSTM layer consisting of a 256 size hidden state. The policy is trained until the policy performance converges on unseen causal structures, and capped at a max of 9 million episodes. In this experiment, we set the horizon $T$ of each episode equal to the number of switches/lights $N$.

## C   CAUSAL INDUCTION QUALITATIVE EXAMPLES

Here we demonstrate an example trajectory and how the causal induction model iteratively builds the causal graph.

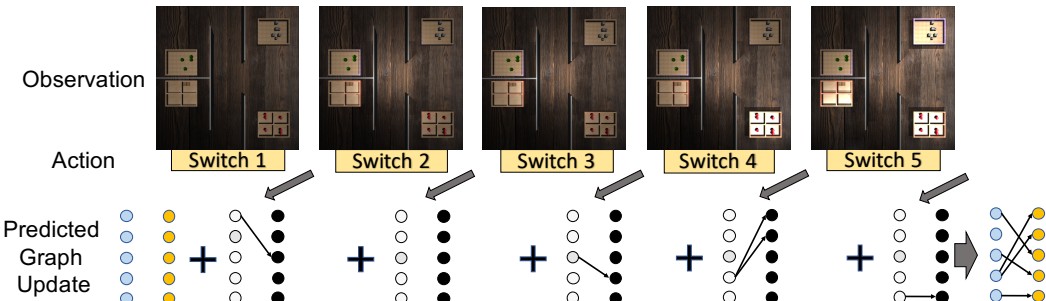

Figure 6: **Sample of Causal Induction**. Here we show an example of our Iterative Causal Induction Model for 5 switches, in the "One-to-Many" case. Given the trajectory of actions and images of the scene, the model needs to reason about which lights were turned on, and how what update this implies in the graph. In this example, the first observed action turns on one of the switches, and the model makes the corresponding update to the graph. The next switch does not change the lighting so the model outputs no update to the graph. The next action sees one light go on, and updates the corresponding switch. The next action turns on two lights, and the graph is updated to reflect this. Lastly, since one light remains unaccounted for, the model knows to add that edge to the graph. Note: The edges and updates are soft updates, but the model learns to predict close to exactly 1 for edges and exactly 0 for non-edges.

