# OpenReview forum: "Causal Induction from Visual Observations for Goal Directed Tasks"
_ICLR.cc/2020/Conference — Reject_

### Official Review · AnonReviewer2 · 2019-10-13
**Official Blind Review #2**

**Rating:** 6

**Review:**

This is a paper about a very interesting topic, involving both learning (in a supervised way) to induce a causal graph and taking advantage of it in a goal-conditioned policy.

This is clearly a timely topic and I loved the motivations of the paper. My main difficulty was with understanding the actual architecture and its motivation, but I believe this is fixable but is a serious impediment to being able to evaluate the paper, as it stands. I was a bit disappointed to see that training is mostly supervised (both providing the ground truth causal graph and an oracle policy as target) but on the other hand it is impressive to obtain these results with raw images as input and the comparative results are good.

First, I would like to better understand the insight behind the architecture of F and several things would need to be clarified to enable reproducibility and making sense of the equations. I would start by suggesting to add an example illustrating why simply seeing a (state,next-state,action) triplet is sufficient to obtain a bit of evidence in favour of a particular edge of the graph. Since this is supervised learning of the causal graph, I imagine that the semantics of the node is predetermined, which is a bit disappointing (but doing otherwise would be understandably much more challenging). Second, I don't understand the structure of the causal graph C. What are the input nodes? action values? action x state cross-product? What are the output nodes? Effect variables? Why would C be NxN and not have different input and output dimensions? All this really needs to be clarified. Based on the 1st eqn of page 4 (PLEASE NUMBER YOUR EQUATIONS!!!) it looks like C is number of actions by number of actions, which does not seem consistent with any reasonable interpretation. The authors should also clarify how R is computed (if it is a straight difference of the encoder output, put up an equation for example) and how delta e is computed.

Then in sec 3.2 the authors talk about a weighted sumn involving selected edges. I imagine this is soft-attenetion but it needs to be clarified with equations and explanations. What is the "content" associated with each edge e which gets averaged in the soft attention?

I found a possibly interesting parallel between the focus of attention on one edge of the graph  at a time (figure 3) and the ideas of the "Consciousness Prior" (Bengio 2017, on arXiv) bottleneck (where only a small tuple of variables, corresponding to an edge here, is considered at each time step in order to reason, plan, decide etc). The fact that in the experiments this attention mechanism helps seems to support the sparsity of dependencies hypothesis underlying the consciousness prior.

My rating is weak reject but I am ready to upgrade with appropriate explanations answering the above questions.

--- post-rebuttal  addition ---

The authors have satisfied most of my concerns and I have upgraded my rating to weak-accept.



**Experience Assessment:**

I have published one or two papers in this area.

**Review Assessment: Checking Correctness Of Derivations And Theory:**

N/A

**Review Assessment: Checking Correctness Of Experiments:**

I assessed the sensibility of the experiments.

**Review Assessment: Thoroughness In Paper Reading:**

I read the paper at least twice and used my best judgement in assessing the paper.

---

> ### Author Response · Authors · 2019-11-11
> **Response to Review #2**
>
> We thank reviewer 2 for their comments and questions. We attempt to clarify them below.
>
> “I would start by suggesting to add an example illustrating why simply seeing a (state,next-state,action) triplet is sufficient to obtain a bit of evidence in favour of a particular edge of the graph.”: Figure 6 in the Appendix includes a qualitative example of the graph induction, including how a specific change in state can map to a predicted edge update to the graph. The core idea is that from state and next-state one can compute the change in state which came from the action, which would imply a certain edge should exist in the graph.
>
> “I imagine that the semantics of the node is predetermined:” Correct, the semantics of the node are predetermined to be one of the switches or one of the lights. The causal induction stage amounts to predicting the directed edges between switch nodes and light nodes from visual observations.
>
> “I don't understand the structure of the causal graph C:” We have restructured section 4.1 to have a paragraph titled “Causal Graph” which describes the causal graph in more depth. We include the following details: First, in the One-to-One, One-to-Many, and Many-to-One settings the graphs are bipartite, consisting of N “cause” nodes corresponding to the switches, and M “effect” nodes corresponding to lights. In practice in our experiments we use M=N, hence the N x N space of mappings between causes and effects. We agree the without this description the fact that the graph is N x N may be confusing, and we have made sure to clarify this in the text. The causal induction stage corresponds to identifying the correct edges out of the possible N x N relationships. In the Masterswitch stage, the graph is tripartite, with an additional mapping between one “master” switch to all other switches.
>
> “The authors should also clarify how R is computed (if it is a straight difference of the encoder output, put up an equation for example) and how delta e is computed.”: Yes, R is computed by computing the straight difference between the encoder output, we have added an equation (Equation 1) for this in Section 3.1. Delta e is the output of a fully connected layer which has shape 1xN, which is a mapping of edge strengths to the N effect variables (lights). Then this is combined with the attention as described in Equation 3 (We have now numbered our equations).
>
> “What is the ‘content’ associated with each edge e which gets averaged in the soft attention?”: For the goal conditioned policy, the soft-attention is over the N “effect” variables (lights), which is applied in a weighted sum resulting in N edge strengths from the N “cause” variables, which is what the policy uses when picking actions. Essentially, this is focusing only on the edges which go into the relevant “effect” variable based on the current state and goal.
>
> “I found a possibly interesting parallel between the focus of attention on one edge of the graph  at a time (figure 3) and the ideas of the ‘Consciousness Prior’”: There certainly are parallels between the proposed ideas in ‘Consciousness Prior’. In particular, the idea that using attention to focus on only relevant parts of data given the current timestep can improve performance is very similar to our proposed approach, and seems to be supported by our empirical findings. We have added citation to this work. Thank you for pointing this out.

---

### Official Review · AnonReviewer1 · 2019-10-28
**Official Blind Review #1**

**Rating:** 3

**Review:**

[Summary]

This paper proposes an interactive agent that tries to infer the underlying causal structure by interacting with the environment; the authors called it "causal induction."  The inferred graph will later help the agent complete goal-directed tasks referred to as a "causal inference" stage. Notably, the agent directly learns from visual inputs.

Both the induction and inference phases heavily rely upon the attention mechanism, which ensures that the agent only focuses on the relevant components of the causal graph. During the induction phase, the agent incrementally updates the predicted causal graph through each interaction using an attention-based edge decoder. During the inference phase, the attention bottleneck also showed to improve the agent's generalization ability.

They have shown that the proposed model outperforms several baselines in a synthetic environment that uses switches to control lights. They have also demonstrated the model's generalization ability by operating on unseen causal graphs and new task goals.


[Major Comments]

My primary concern about this work is the scope of its applicability.

For the causal induction phase, the proposed method makes a strong assumption that it can access the ground truth causal relationship during training. The authors can directly read this information from the synthetic environments used in this paper, yet, in more complex real-world situations, we might not know the underlying causal structure for supervised training the induction model.

For learning the goal-conditioned policies, the authors also assume that they have access to the ground truth causal graph. They use this information to generate the expert demonstrations, which, I presume, are deterministic and unimodal (correct me if I'm wrong). In the real world, a human may be able to infer the underlying causal structure from the observations by interacting with the environment and provide the demonstration data. However, the demonstration may be noisy or form a multi-modal distribution. While I agree that learning from demonstration is an effective way of guiding the learning of the policy, I'm not sure if the method can generalize to more realistic scenarios.

For inferring the causal graph, the authors also assume that they know the "cause" set and the "effect" set, which is already a DAG by construction. Instead of inferring the direction of the edge, they are solving an easier problem of deciding whether a directed edge between a "cause" node and an "effect" node exists or not. The assumption on the graph structure also limits the method's applicability, as, in the real world, the direction of the edge is not always known in advance. Smoking may cause lung cancer, but it is possible that lung cancer may make people smoke more.

I feel this paper makes strong assumptions on both the induction and inference stages, as well as the structure of the causal graph, which greatly limits the applicability of the approach.

[Detailed Comments]

I also have a few questions regarding the details of this paper.

In Section 3.1, the authors said that "N is the number of actions in the environments," which is a bit confusing. Before this point, the authors did not discuss the relationship between the size of the graph and the size of the action set. Only until Section 3.2 did I realize that N is the number of both "cause" set and "effect" set. It would be better to discuss the size of the graph and the action set at an earlier position.

How is the expert planner implemented? Is the expert's policy deterministic and unimodal? I feel this is an important detail to include.

It is related to the previous question. What will happen if we learn the goal-conditional policy from scratch? How much does imitation learning help with policy learning? Again, the assumption that we know the ground truth causal graph may not be feasible in the real world.

In Section 3.3, the authors said that "the expert's action is added to the memory of the policy." However, the authors also noted that "the policy has no memory" in Section 3.2, which seems to contradict each other. Does the "memory" mean replay buffer in Section 3.3?

Does the number of switches fixed across all environments and always the same as the number of lights? Also, are all the lights mounted in the same location? I'm wondering if the model is invariant to the order of the cause nodes and effects nodes in the graph, and can it generalize to larger environments, more lights, and different room configurations.

In Figure 4, TCIN seems to have the best performance in the "Masterswitch" environment when there are 500 seen causal structures. What might be the reason?

**Experience Assessment:**

I have published one or two papers in this area.

**Review Assessment: Checking Correctness Of Derivations And Theory:**

I assessed the sensibility of the derivations and theory.

**Review Assessment: Checking Correctness Of Experiments:**

I assessed the sensibility of the experiments.

**Review Assessment: Thoroughness In Paper Reading:**

I read the paper at least twice and used my best judgement in assessing the paper.

---

> ### Author Response · Authors · 2019-11-11
> **Response to Review #1**
>
> We thank reviewer 1 for their thoughtful comments. We address each comment below:
>
> Scope of applicability: It is correct that we assume access to the ground-truth causal structure in the training of our final model. We also experimented with different setups, where we trained causal induction without ground-truth causal graph in the “Memory” baseline and trained policy without ground-truth causal graph in the “Memory (RL/Low Dim)” baseline. However, both baselines failed to match the performance of our final model (see Figure 5). It implies that the use of ground-truth causal graph is crucial for learning efficiency and high performance of our model. In more realistic scenarios, we would have to investigate more effective reinforcement learning algorithms as a replacement to imitation learning to reduce the reliance on expert demonstrations.
>
> In addition, it is true that we assume that the graph nodes are known and focus on predicting the directed edges. Under this assumption, our core technical contribution is using attention mechanisms to selectively read and update the causal graph. We plan to explore the joint learning of nodes and edges of a causal graph in future work.
>
> “Only until Section 3.2 did I realize that N is the number of both ‘cause’ set and ‘effect’ set”: Thanks for pointing this out. We have also added a brief description of the graphs we consider in the problem statement, as well as a more in-depth description of the causal graphs in Section 4.1.
>
> “How is the expert planner implemented?”: Using the goal state and current state, the expert computes the necessary change in state, and using the ground truth causal graph plans a sequence of actions which would achieve this change. Hence the expert policy is deterministic and unimodal - however note we train using DAgger with epsilon greedy exploration, so rather than receiving demonstrations, we query the expert online while training the policy. We have added this clarification to Section 3.3.
>
> “What will happen if we learn the goal-conditional policy from scratch?”: We tried to train the policy with reinforcement learning (RL) on image inputs, but it did not converge. Therefore, we trained an RL baseline which learned a single policy using memory and ground-truth low-dimensional states (no images) called “Memory (RL/Low dim)”, and its performance was still low (See Figure 5). This suggests that imitation learning is crucial for handling high-dimensional observations and improving sample efficiency in our problem setup.
>
> “Does the ‘memory’ mean replay buffer in Section 3.3?”: Correct, the experts action is added to the policy replay buffer. When we say the policy has no memory we mean it is a feed-forward network. We have corrected this term in Section 3.3.
>
> “Is the number of switches fixed across all environments and always the same as the number of lights? Also, are all the lights mounted in the same location?”: Our experiments look at environments with 5 switches 5 lights or 7 switches 7 lights. And yes for one environment the position of the lights are the same, but the mapping between switches and lights changes. So one trained model is not invariant to the ordering of cause/effect nodes in the graph, but the method in general should extend to larger environments/lights/room configurations (assuming you train one model per configuration).
>
> “In Figure 4, TCIN seems to have the best performance in the "Masterswitch" environment when there are 500 seen causal structures. What might be the reason?”: This is likely because with 500 training structures the inductive bias of attention is no longer necessary to successfully generalize, and the end-to-end method with temporal convolutions can generalize well.

---

### Official Review · AnonReviewer3 · 2019-10-28
**Official Blind Review #3**

**Rating:** 6

**Review:**

The authors describe a method for endowing an artificial agent with causal reasoning  when completing goal-directed tasks. Causal reasoning knowledge is encoded in a directed acyclic bipartite graph or slightly more complicated "master switch" variation.

Essentially the method is:
  - train a model F to predict causal graphs (for which we have ground truth data) from trajectories generated with a heuristic,
  - train an policy \pi_G attending over the causal graph to solving tasks.

At inference time F an \pi_G are frozen and are evaluated on similar tasks to the ones in training, where however the (unseen) causal graph is new. Thus the evaluation is in a meta-learning scenario, where the model has to learn how to learn to solve tasks.

The experimental setting is an agent controlling 5 or 7 switches in a simulated environment and observing 32x32x3 images of the environment.

The authors report significant improvements over an existing baseline (Dasgupta et al. (2019)). The key to the improvements is in the iterative prediction of the causal graph (for F) and in having attention over the causal graph (in \pi_G).

The paper is describes a very nice new method and makes interesting points about the role that causal reasoning can play in modeling agents in goal-directed tasks. The paper is strengthened by improvements on existing baselines based on their intuitions.

Questions:

* The authors claim that constructing an explicit causal structure, instead of a latent feature encoding, leads to better generalization in “long-horizon” tasks -- but they seem to only test on one task, fairly limited in complexity.

* On page 1, the authors claim that empirical evidence suggests the lack of correct causal modeling is an important factor for lack of generalization, generation of unrealistic captions, and difficulties in transfer learning. This seems to be a bit of an overreach. It’s possible that many empirical problems could be solved to a large degree by advances in machine learning without having to resort to explicit causal modeling.

* On page 2, is the reward function r the same for all MDPs?

* Could it be said from the start that pi_I is heuristic and described in section 4.1?

* There seem to be strong assumptions on the structure of C that are only stated late in the paper. C is initially presented as an arbitrary acyclic graph, but from Figure 2 it appears to instead be a bipartite graph with N source nodes and N target nodes, and potential connections from any source node to any target node, where N is the number of actions. This should be explained early on. The “master switch” variation does not seem to exactly fit the mathematical descriptions from the “Methods” section.

* \hat{C}_H does not seem to be defined except inside Figure 2. It should be defined in the text.


**Experience Assessment:**

I have published one or two papers in this area.

**Review Assessment: Checking Correctness Of Derivations And Theory:**

I carefully checked the derivations and theory.

**Review Assessment: Checking Correctness Of Experiments:**

I assessed the sensibility of the experiments.

**Review Assessment: Thoroughness In Paper Reading:**

I read the paper thoroughly.

---

> ### Author Response · Authors · 2019-11-11
> **Response to Review #3**
>
> We thank reviewer 3 for their comments. We address each comment below:
>
> “Long-horizon tasks”: By long-horizon we meant that our causal inference task consists of sequential decision making, as opposed to the one-step querying of previous work (Dasgupta et al. 2019). We have rephrased to "multi-step" tasks in the revision.
>
> “It’s possible that many empirical problems could be solved to a large degree by advances in machine learning without having to resort to explicit causal modeling”: Absolutely. Lack of causal reasoning is one of the hypotheses for weak generalization. Our method provided one way to address this problem by constructing an explicit causal model; it could also be addressed by other advances in machine learning.
>
> “Is the reward function r the same for all MDPs?”: Yes, the reward function is the difference between the current state and goal state across all MDPs.
>
> “Could it be said from the start that pi_I is heuristic and described in section 4.1?”: Yes we have added this when we first mention pi_I in the problem statement.
>
> “There seem to be strong assumptions on the structure of C that are only stated late in the paper.”: Thank you for pointing this out. For clarity, we have updated the problem statement to mention that in this work we focus on bipartite graphs, and also expand on the details of C in Section 4.1.
>
> “\hat{C}_H does not seem to be defined except inside Figure 2.”: \hat{C}_i, corresponds to the estimate of \hat{C} at the ith iteration. We have added a description of \hat{C}_i in the caption to clarify this - thank you for pointing this out.

---

### Decision · Program_Chairs · 2019-12-19

**Decision:**

Reject

**Comment:**

The submission presents an approach to uncovering causal relations in an environment via interaction. The topic is interesting and the work is timely. However, the experimental setting is quite simplistic and the approach makes strong assumptions that limit its applicability. The reviewers are split. R2 raised their rating from 3 to 6 following the authors' responses and revision, but R1 maintained their rating of 3 and posted a response that justifies this position. In light of the limitations of the work, the AC recommends against accepting the submission.